# Will the Real Linda Please Stand Up...to Large Language Models? Examining the Representativeness Heuristic in LLMs

**Pengda Wang**[*], **Zilin Xiao**[*], **Hanjie Chen, Frederick L. Oswald**
Rice University
{pw32,zilin,hc86,fo3}@rice.edu

## Abstract

Although large language models (LLMs) have demonstrated remarkable proficiency in modeling text and generating human-like text, they may exhibit biases acquired from training data in doing so. Specifically, LLMs may be susceptible to a common cognitive trap in human decision-making called the *representativeness heuristic*. This is a concept in psychology that refers to judging the likelihood of an event based on how closely it resembles a well-known prototype or typical example, versus considering broader facts or statistical evidence. This research investigates the impact of the representativeness heuristic on LLM reasoning. We created REHEAT (Representativeness Heuristic AI Testing), a dataset containing a series of problems spanning six common types of representativeness heuristics. Experiments reveal that four LLMs applied to REHEAT all exhibited representativeness heuristic biases. We further identify that the model's reasoning steps are often incorrectly based on a stereotype rather than on the problem's description. Interestingly, the performance improves when adding a hint in the prompt to remind the model to use its knowledge. This suggests the uniqueness of the representativeness heuristic compared to traditional biases. It can occur even when LLMs possess the correct knowledge while falling into a cognitive trap. This highlights the importance of future research focusing on the representativeness heuristic in model reasoning and decision-making and on developing solutions to address it.

## 1 Introduction

*"Linda is 31 years old, single, outspoken, and very bright. She majored in philosophy. As a student, she was deeply concerned with issues of discrimination and social justice and also participated in anti-nuclear demonstrations."*

**We have seven statements below:** *[1] Linda is a teacher in elementary school. [2] Linda works in a bookstore and takes Yoga classes. [3] Linda is a psychiatric social worker. [4] Linda is a member of the League of Women Voters. [5] Linda is a bank teller. [6] Linda is an insurance salesperson. [7] Linda is a bank teller and is active in the feminist movement.*

**Question:** *Rank the seven statements associated with each description by the degree to which Linda resembles the typical member of that class.*

- Tversky & Kahneman (1983)

This is one experiment known as the "Linda problem," devised by Tversky and Kahneman; results demonstrate how respondents are influenced by specific descriptions, such as being `deeply concerned with issues of discrimination and social justice and also participating in anti-nuclear demonstrations` to rank [7] higher than [5]. However, [7] combines [5] and an additional event. This means that, from a statistical perspective, [5] is more likely to occur than [7] because it is more general and less restrictive.

---

[*]Equal contribution

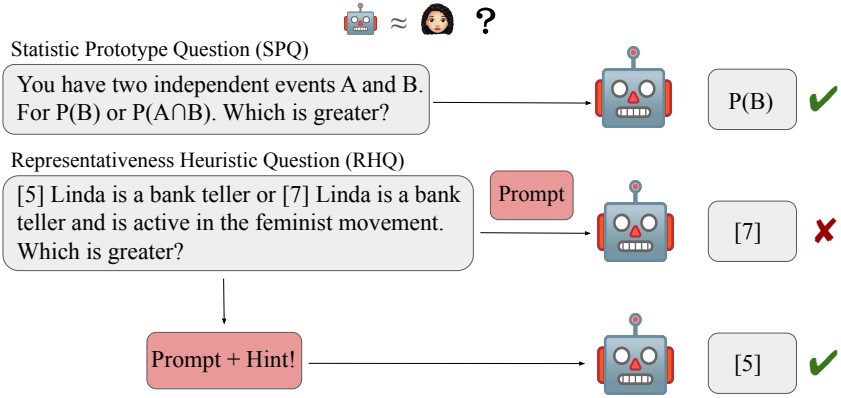

Figure 1: Illustration of the representativeness heuristic problem. The model possesses the knowledge to answer the statistical prototype question, yet fails to use it to solve the representativeness heuristic question. Providing an appropriate hint in the prompt can guide the model in making a correct prediction.

Kahneman & Tversky (1973) introduced this phenomenon as the "representativeness heuristic," which involves estimating the likelihood of an event by comparing it to an existing prototype in our minds. It offers a convenient shortcut in decision-making by aligning with intuitive thinking, leading people to rely on it frequently. Generally, this heuristic is quite beneficial, as it simplifies complex judgments. However, it is important to recognize that it can also result in significant decision-making errors, given that people are prone to assessing the likelihood of an object's category membership based on superficial similarities, while neglecting actual statistical evidence. For example, when they lack the benefit of individuating information about a person, decision-makers might categorize individuals based on their physical appearance, observation of limited behaviors, or superficial background descriptions, leading to skewed perceptions and decisions. This type of stereotyping phenomenon is durable, being widespread across cultures and over time (Spencer et al., 2016).

Because LLMs are trained on real-world data and instructed to emulate human behavior, they may capture the representativeness heuristic. Previous work has mainly focused on the biases within training data (Bender et al., 2021; Bolukbasi et al., 2016; Garg et al., 2018; Sheng et al., 2019; Zhao et al., 2019). These biases often stem from data distributions that do not reflect proportions that would drive decision-making that is unbiased, given a researcher's operational definition of bias. In contrast, the representativeness heuristic represents a type of cognitive bias that has yet to be thoroughly investigated in LLMs. It is unique in potentially leading models to make mistakes, even when they possess the knowledge necessary to solve the problem. As illustrated in Figure 1, the model is able to answer the Statistical Prototype Question (SPQ), yet it tends to fail to answer the Representativeness Heuristic Question (RHQ). The SPQ and RHQ are intrinsically equivalent, where SPQ is expressed statistically, and RHQ expresses the same statistical logic verbally in a scenario. This indicates that LLMs can engage in erroneous cognitive reasoning, even with accurate knowledge of statistical probabilities. Interestingly, providing an appropriate hint can prompt an LLM to use its knowledge to make correct predictions more frequently. This indicates that the representativeness heuristic can block the model from following a correct reasoning path; instead, it relies on the data that captures the human tendency to use cognitive shortcuts to make decisions.

To investigate the representativeness heuristic in LLMs, we construct a dataset[1] , REHEAT (Representativeness Heuristic AI Testing), which contains 202 RHQs that span six types of representativeness heuristics: Base Rate Fallacy, Conjunction Fallacy, Disjunction Fallacy, Insensitivity to Sample Size, Misconceptions of Chance, and Regression Fallacy. The ques-

---

[1]Code and dataset will be available at https://github.com/MrZilinXiao/LLMHeuristicReHEAT.

tions we designed are adapted from those used in prior investigations into heuristics within the field of psychology (Bar-Hillel & Neter, 1993; Kahneman & Tversky, 1973; Kahneman et al., 1982; Tversky & Kahneman, 1974; 1983). To the best of our knowledge, our dataset is the first to offer extensive and comprehensive coverage of RHQs, enabling exploration of LLMs' capabilities in countering the representative heuristic cognitive bias.

We evaluate four LLMs: GPT-3.5 (Ouyang et al., 2022), GPT-4 (OpenAI, 2023), PaLM 2 (Anil et al., 2023), and LLaMA 2 (Touvron et al., 2023), on REHEAT using different prompts. Our findings indicate that these LLMs exhibit behaviors that closely mirror human heuristic behavior. Additionally, advanced prompting techniques such as chain of thought (CoT) (Wei et al., 2022), in-context learning (Brown et al., 2020), and self-consistency (Huang et al., 2022) prompting, offer marginal improvements. Nevertheless, when explicitly prompted to recall its knowledge, the model shows an improvement in performance. This underscores the significance of future research to address the representativeness heuristic, guiding LLMs toward correct reasoning and decision-making.

## 2 Related Work

Social biases in natural language processing (NLP) systems and related data have been studied with respect to their fairness, inclusivity, and accuracy (Hutchinson et al., 2020; Maass, 1999; Zhao et al., 2018). For example, Bolukbasi et al. (2016); Garg et al. (2018) are among the pioneers in demonstrating gender-related associations in word embeddings that might reflect and perpetuate stereotypes. Caliskan et al. (2017) concludes standard machine learning methods for NLP could acquire societal biases from textual data. Some research has expanded our understanding of where NLP systems might acquire subgroup associations within data, and potential bias, including those from data collection (Bender & Friedman, 2018), annotation processes (Gebru et al., 2018), and model architecture choices (Zhao et al., 2017). To date, numerous efforts have been made to mitigate social biases in systems through a variety of methods, including data augmentation (Lu et al., 2018), changes in model architecture (Liang et al., 2020), and training objectives (Liu et al., 2021; Romanov et al., 2019).

In a similar vein, although recent advancements in LLMs are exciting, researchers are concerned about whether LLMs inherit social biases from the trillions of tokens they have been trained on. Weidinger et al. (2022) provides a comprehensive taxonomy of social risks within LLMs. Although the research community has documented numerous social biases in LLMs (Ferrara, 2023; Mei et al., 2023), few LLM researchers have examined these biases from the perspective of the psychology of human decision-making. Thus, in the present work, we study the bias issue in LLMs from the new angle of the aforementioned representative heuristic, a concept originating in psychology (Kahneman & Tversky, 1973; Kahneman et al., 1982; Tversky & Kahneman, 1974; 1983).

Contrary to the very recent work of Suri et al. (2024), which explores the decision-making heuristics of GPT-3.5, our current research emphasizes a more in-depth and comprehensive exploration of the application of the representativeness heuristic within LLMs. For example, we have compiled a dataset encompassing a much larger number of questions (a total of 202 compared to 9) and we have benchmarked performance across a wider range of LLMs based on diverse prompting strategies.

## 3 Representativeness Heuristic

Drawing from prior research, we organize our research around a framework that categorizes the representativeness heuristic into six types (Kahneman & Tversky, 1973; Kahneman et al., 1982; Tversky & Kahneman, 1974; 1983). These categories vary in their fundamental logical approach and their impact on decision-making processes.

**Base Rate Fallacy** occurs when individuals overlook or insufficiently account for the population base rate of an event (i.e., its overall prevalence or frequency) in favor of specific instances or recent information. Figure 2a presents an example where $P(B)$ represents the

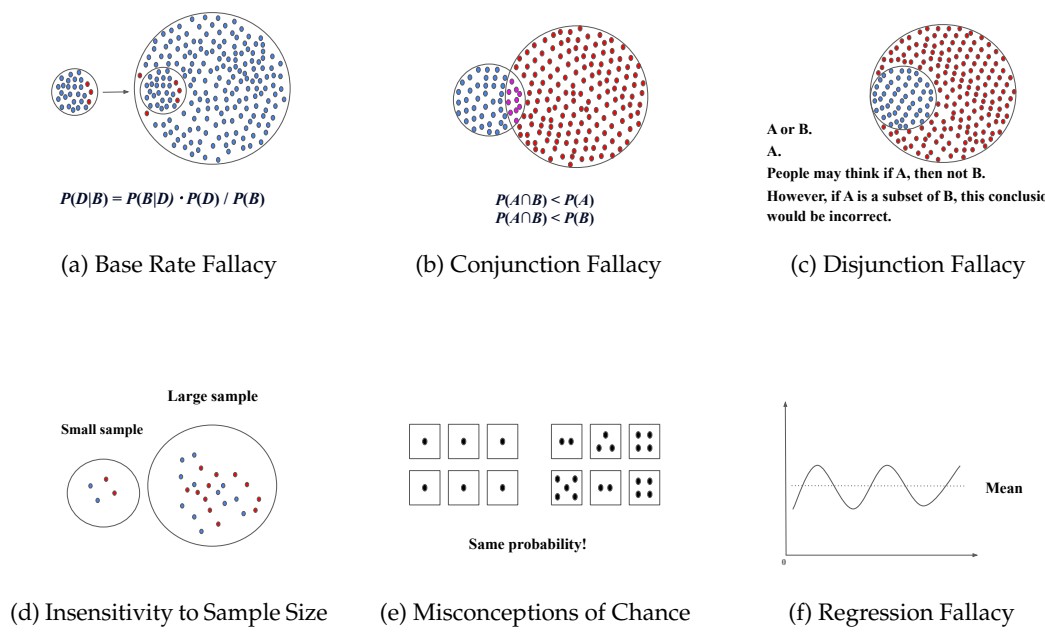

(a) Base Rate Fallacy    (b) Conjunction Fallacy    (c) Disjunction Fallacy

(d) Insensitivity to Sample Size    (e) Misconceptions of Chance    (f) Regression Fallacy

Figure 2: Illustrations on six types of representativeness heuristic

proportion of individuals with a symptom (all blue points within the large circle), $P(D)$ denotes the rate of illness (all points within the small circle), and $P(B|D)$ indicates the proportion of the sick who have the symptom (all blue points within the small circle). $P(D|B)$ represents the illness rate among those with the symptom. Most people would assume that because $P(B|D)$ is high, $P(D|B)$ would also be high. Yet, according to Bayes' theorem,

$$P(D|B) = \frac{P(B|D) \cdot P(D)}{P(B)},$$

meaning $P(D|B)$ is greatly influenced by the base rates of $P(B)$ and $P(D)$, showing the importance of considering general prevalence in evaluating specific probabilities. An example question can be seen in Table 18 of the Appendix.

**Conjunction Fallacy** occurs when people mistakenly believe that the chance of two events happening together is greater than the chance of either event happening alone. See Figure 2b, consider the example at the beginning of the article. $P(A)$ represents the probability that the person is a bank teller (all points within the small circle, less relevant to the description). $P(B)$ is the probability that the person is active in the feminist movement (all points within the large circle, more relevant to the description). $P(A \cap B)$ the probability that the person is both a bank teller and active in the feminist movement (all purple points). $P(A \cap B)$ will always be no larger than $P(A)$ and $P(B)$, no matter which one is closer to the description. An example question can be seen in Table 19 of the Appendix.

**Disjunction Fallacy** occurs when people incorrectly judge the probability of a broader category to be smaller than one of its specific components. In Figure 2c, we can imagine the small circle representing ice cream and the large circle representing frozen food. Since ice cream is a subset of frozen food, the probability of frozen food is higher than ice cream. However, when people talk about summer refreshments, they often think of ice cream rather than frozen food. This choice illustrates a common tendency to differentiate specific items from their general classifications based on contextual associations. An example question can be seen in Table 20 of the Appendix.

**Insensitivity to Sample Size** occurs when people underestimate how important sample size is in data evaluation, potentially leading to incorrect conclusions. Figure 2d presents

an example where a small group (small circle) and a large group (large circle) both have 50% blue dots and 50% red dots (2:2 and 12:12). If we add one red dot to each sample, then the ratio of blue to red becomes 40% and 60% (2:3) in the small group, versus 48% and 52% (12:13) in the large group. It should be recognized that smaller groups are more prone to skewed outcomes because even small changes have a larger impact on the overall dynamics of a small group. An example question can be seen in Table 21 of the Appendix.

**Misconceptions of Chance** involve misunderstanding how randomness works, especially thinking that past outcomes will affect future outcomes in cases where the outcomes are, in fact, independent. For example, Figure 2e presents a dice-rolling example. People wrongly believe that if a specific outcome has occurred frequently, it is less likely to happen again shortly, or vice versa if it has occurred rarely. However, in truly random events, such as rolling a fair die, the probability of any given outcome (1-6) remains constant at $\frac{1}{6}$, unaffected by the sequence of previous results. An example question can be seen in Table 22 of the Appendix.

**Regression Fallacy** occurs when individuals overlook the natural tendency for extreme situations to move back towards the average (regress to the mean) and instead, erroneously assign this regression to a particular cause, see Figure 2f. For instance, if an athlete shows a lackluster performance following a perfect game, it might be incorrectly ascribed to external factors, neglecting the likelihood of natural variance in performance. An example question can be seen in Table 23 of the Appendix.

## 4 Dataset and Experimental Setup

We begin by constructing a dataset intended to reflect a diverse array of questions and scenarios (§4.1). Following this, we introduce the models and prompting strategies (§4.2), along with the evaluation methods (§4.3).

### 4.1 Data Collection

Our main resource for creating test questions on representativeness heuristic comes from academic research by Kahneman and Tversky (Kahneman & Tversky, 1973; Kahneman et al., 1982; Tversky & Kahneman, 1974; 1983). Their work introduced various question types and insights into the design of cognitive heuristic measures. Building on their foundation, we design questions that extensively explore the representativeness heuristic in LLMs. Specifically, our test set contains 49 questions drawn directly from previous research and 153 new questions that have been carefully adapted. The reason for creating these new questions is to provide diverse contexts and achieve stronger external validity. In creating our adapted items that change the situation and context of the original items, we paid special attention to retaining the essence of the representativeness heuristic being tested.

The total 202 examples are all in English. As shown in Table 1, each example is composed of the following characteristics:

1. **Query**: Provide the question's description and requirements.
2. **Type**: Indicate the question's category, such as: *Disjunction Fallacy*.
3. **Feature**: Highlight unique aspects, such as *choose one*, or the question's source, such as: *original problem in Bar-Hillel & Neter (1993)*.
4. **Ground Truth**: Represents the standard answer.
5. **Human Response**: Document the outcomes of human responses from academic research when available, or denoting *N/A* otherwise.

### 4.2 Models and Prompting Strategies

We investigate four LLMs, encompassing both closed-source and open-source models: GPT-3.5 (gpt-3.5-turbo-0613) (Ouyang et al., 2022), GPT-4 (gpt-4-0613) (OpenAI, 2023), PaLM

| FIELD | DESCRIPTION |
|---|---|
| Query | Danielle is sensitive and introspective. In high school, she wrote poetry secretly. Did her military service as a teacher. Though beautiful, she has little social life since she prefers to spend her time reading quietly at home rather than partying. What is she most likely to study? Choose one answer that follows: [1] Literature, [2] Humanities, [3] Physics, [4] Natural science. |
| Type | Disjunction Fallacy |
| Feature | Choose one, original problem in Bar-Hillel & Neter (1993) |
| Ground Truth | [2] Humanities |
| Human Behavior | Disjunction Fallacy Rate 50% - 56% |

Table 1: Exemplar in REHEAT

2 (`chat-bison-001`) (Anil et al., 2023), and LLaMA 2 (`llama-2-70b-chat`) (Touvron et al., 2023). We apply four different prompting strategies to each of them to generate answers.

**Standard**: We ask the model to directly answer the query without explicit reasoning instructions with a greedy decoding method.

**Zero-shot Chain-of-Thoughts (CoT)**: We first ask the model to generate its reasoning with an instruction[2], then direct it to answer the query with the context of reasoning steps[3]. This two-step CoT strategy is developed based on CoT (Wei et al., 2022) and its zero-shot variation (Kojima et al., 2022).

**Self-Consistency**: We prompt the model to generate ten answers with a temperature sampling parameter of $T = 0.7$, using majority voting to finalize the model decision via diverse reasoning paths. This prompting strategy is also known as self-consistency prompting (Wang et al., 2023).

**Few-shot In-Context Learning (ICL)**: The model will be prompted with a few selected examples from the same category of representativeness heuristic so that the model can learn a task from demonstrations. Samples used as exemplars will not participate in accuracy calculations. This ability to learn from context is known as in-context learning (Brown et al., 2020).

### 4.3 Evaluation Methods

**Automatic evaluation:** All query questions in REHEAT are either in multiple-choice or ranking format. We adopt precise prompt templates in Table 14 of the Appendix to instruct models to generate responses. For a multiple-choice question, a response is deemed correct if and only if it contains the ground-truth option. For a ranking question, the exact match between relative permutations of response options and ground-truth options qualifies for a true positive model prediction.

**Human evaluation:** In addition, one of the present authors with expertise in psychology conducted the human evaluation to assess the output reasoning steps from zero-shot CoT prompting. We report the proportions of the four possible LLM outcomes: both reasoning and prediction are correct; the reasoning is correct, but the prediction is incorrect; the

---

[2]Step 1: *Let's think step by step, but don't give the answer directly.*
[3]Step 2: *Therefore, the answer is.*

prediction is correct, but the reasoning is incorrect; both reasoning and prediction are incorrect.

# 5 Experimental Results

In addition to evaluating the models' performance on the REHEAT dataset and analyzing their reasoning abilities (§5.1; §5.2), we further investigated the potential performance boost that could be achieved by providing hints to the models (§5.4), as well as how situational similarity influences model performance (§5.3).

## 5.1 LLMs Exhibit the Representativeness Heuristic

We report the model's performance on the REHEAT dataset with different prompting strategies, shown in Table 2. Although GPT-3.5 shows the strongest performance in standard prompting, which we will elaborate on later in §5.2, the advanced prompting methods of zero-shot CoT, self-consistency, and one-shot ICL have a negative impact on its performance. GPT-4 and LLaMA-2-70B benefit from these prompting strategies with noticeable growth in prediction accuracy, but it is still not substantial. LLaMA2-70B demonstrates the most significant improvement (+14.5%) with one-shot ICL prompting, possibly due to the standard prompting on LLaMA2-70B results in response formatting issues. PaLM 2 is the least effective LLM, with its performance lagging behind other models using standard prompts, despite applying prompt strategies.

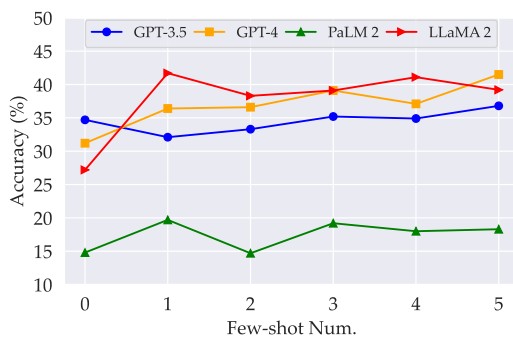

Figure 3: In-context learning accuracy of four selected LLMs on REHEAT with few-shot demonstrations.

**How Much Can In-Context Learning Help?** As shown in Table 2, one-shot ICL performs the best among various prompting strategies. This has piqued our curiosity: Could providing more examples further improve performance?

To explore whether LLMs can acquire knowledge related to the representativeness heuristic through ICL, we report the $k$-shot performance of four types of LLMs in Figure 3. From $k = 0$ to 5, both GPT-4 and LLaMA-2 exhibit a noticeable improvement in accuracy from adding an example in context. However, such an uptrend becomes saturated and begins to fluctuate with further increases in the number of shots. In contrast, GPT-3.5 and PaLM 2 do not display a clear pattern of improvement with the addition of more in-context examples, indicating a weak or non-existent correlation between the number of exemplars and accuracy.

**Error Type Analysis** We also check each type of representativeness heuristic (Table 7 - 10 of the Appendix). We have observed that in most cases, most models perform relatively poorly when dealing with questions of Conjunction Fallacy and Disjunction Fallacy. Compared to other types of questions, such as Base Rate Fallacy, the challenge with Conjunction Fallacy and Disjunction Fallacy lies in the fact that the statistical reasoning required is embedded within the connotations and combinations of texts (e.g., `South Africa is a subset of Africa`) rather than being directly indicated by expressions like `large sample size` or `far more than`.

**Do LLMs Possess Statistical Knowledge?** We examine LLMs' knowledge of statistical principles in answering SPQs. The four models possess statistical knowledge, demonstrating a comprehensive understanding across all categories of statistical principles (Table 5 of the Appendix; SPQs example in Table 15 of the Appendix). This significantly differs from our

| Model | Strategy | | | |
|---|---|---|---|---|
| | Standard | Zero-shot CoT | Self-Consistency | One-Shot ICL |
| GPT-3.5 | 34.7 | 30.7 (-4.0) | 35.6 (+0.9) | 32.1 (-2.6) |
| GPT-4 | 31.2 | 35.1 (+3.9) | 34.2 (+3.0) | 36.4 (+5.2) |
| LLaMA2-70B | 27.2 | 29.2 (+2.0) | 29.2 (+2.0) | 41.7 (+14.5) |
| PaLM 2 | 14.8 | 12.4 (-2.4) | 20.8 (+6.0) | 19.7 (+4.9) |

Table 2: Model prediction accuracy (%) on REHEAT. The relative performance of different prompt strategies with respect to standard prompting is indicated with script-size fonts.

| Type | | Model | | | |
|---|---|---|---|---|---|
| | Human | GPT-3.5 | GPT-4 | LLaMA2-70B | PaLM 2 |
| Base Rate Fallacy | 27.7 | 31.9 (+4.2) | 34.0 (+6.3) | 23.4 (-4.3) | 19.1 (-8.6) |
| Conjunction Fallacy | 24.4 | 31.1 (+6.7) | 20.0 (-4.4) | 20.0 (-4.4) | 2.2 (-22.2) |
| Disjunction Fallacy | 12.5 | 22.9 (+10.4) | 6.3 (-6.2) | 10.4 (-2.1) | 2.3 (-10.2) |
| Insensitivity to Sample Size | 33.3 | 45.1 (+11.8) | 54.9 (+21.6) | 43.1 (+9.8) | 22.0 (-11.3) |
| Misconceptions of Chance | 50.0 | 50.0 (=) | 50.0 (=) | 75.0 (+25.0) | 50.0 (=) |
| Regression Fallacy | 85.7 | 71.4 (-14.3) | 71.4 (-14.3) | 71.4 (-14.3) | 71.4 (-14.3) |
| Average | 27.2 | 34.7 (+7.5) | 31.2 (+4.0) | 27.2 (=) | 14.8 (-12.4) |

Table 3: Human performance on REHEAT. A comparison with standard results from LMs.

observations on the accuracy of the four models' performance on RHQs, indicating that the models indeed made errors related to the representativeness heuristic.

**Human Performance on REHEAT**   We also present human performance on REHEAT in Table 3. Through the Prolific platform, we recruited 153 participants (66 males, 86 females, and 1 other) to answer questions. The participants' ages ranged from 19 to 77 years (M = 37.2, SD = 12.6). The racial distribution of the participants was as follows: 70.6% White, 9.8% African American, 7.8% Asian, and 11.8% other. Each participant was asked to respond to 3-4 different representative heuristic questions, providing three responses for each REHEAT question. We used majority voting to finalize the human decision. It can be observed that the human results closely matched the LLaMA2-70B result, while the GPT series performed slightly better than the human results. This aligns with the findings of Bar-Hillel & Neter (1993), Kahneman & Tversky (1973), Kahneman et al. (1982), Tversky & Kahneman (1974), and Tversky & Kahneman (1983), who reported that human accuracy across various tasks ranges from 10% to 50%.

Overall, LLMs have demonstrated a representativeness heuristic bias similar to humans. Example questions and model answers are in Table 18 - 23 of the Appendix.

## 5.2   Discrepancies in Model Reasoning: Beyond Predictions

While testing the REHEAT dataset, we also conducted human evaluations on the reasoning steps produced by the GPT-4 and LLaMA2-70B models under CoT prompting. As before, we form four possible combinations of outcomes, as shown in Figure 4 and Table 11 of the Appendix.

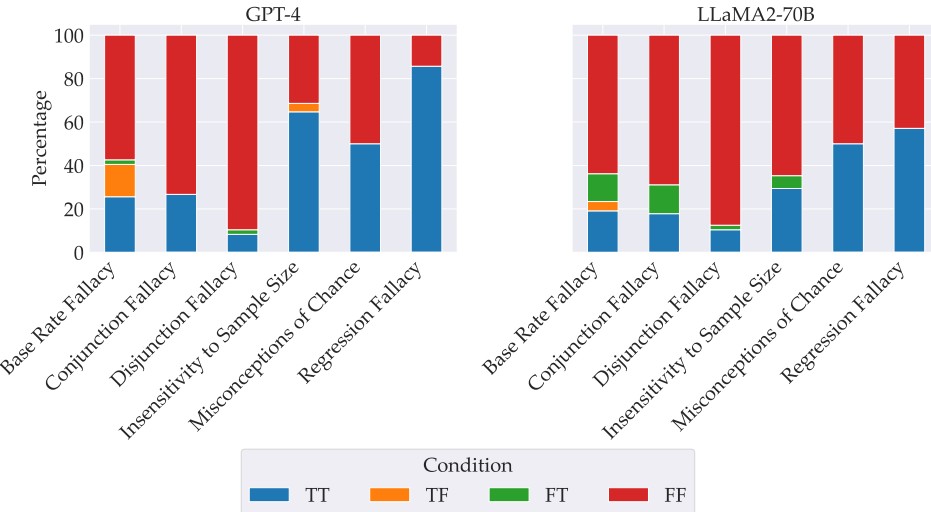

Figure 4: The combination of reasoning and outcome correctness for GPT-4 and LLaMA2-70B on the REHEAT via CoT prompts. Detailed Table 11 in Appendix.

Generally, GPT-4 outperformed LLaMA2-70B in all types of RHQs. However, this improvement is accompanied by a side effect: an increased proportion of instances where GPT-4's reasoning process is accurate, but the outcome is incorrect. This primarily occurs because when faced with ambiguous questions, GPT-4 often preferred to express the need for additional information before making a decision. This is also why GPT-4 accuracy is slightly lower than GPT-3.5 under some prompts. Although we mark such cautious answers as incorrect outcomes—because they did not adhere to the directive of providing a definitive answer—cautious answers may remain beneficial in practical settings. Conversely, LLaMA2-70B more frequently produces correct outcomes from incorrect reasoning processes; the reasons are often incorrectly based on a stereotype rather than the problem's description.

We have conducted case studies on some interesting reasoning, we include the case study in Appendix A. In summary, our findings indicate that LLMs are influenced by semantic content in their reasoning processes, aligning with results from previous research (Acerbi & Stubbersfield, 2023; Dasgupta et al., 2022). Specifically, when the semantic content of tasks was clear and did not contain sensitive content (e.g. gender, race), the models' responses were better.

## 5.3 Improving Performance by Hinting LLMs to Use Their Knowledge

How can the model's cognitive process be put back on track? We test whether LLMs show enhanced performance with prompts that hint for them to use their existing knowledge. We tested two types of hints: one general and the other based on more detailed cues for each representativeness heuristic type (Table 17 of the Appendix). These prompts aimed to hint at the model to recall the knowledge it possessed. Results are presented in Table 4. We also provide detailed results for each type of representativeness heuristic in Table 12 and Table 13 of the Appendix. We found that both types of hints provided a noticeable improvement for most models, with the specific type of hints providing a more significant boost in performance compared to the general prompts.

## 5.4 The Impact of Situational Similarity on Model Performance

After observing the performance differences between the model on statistical SPQs and on the contextual RHQ counterparts found in REHEAT, we delve further into the middle of these two questions - Intermediate Questions (IQs) (Table 16 of the Appendix). This type of question integrates specific situational contexts and statistical data, making it necessary to consider both concrete data and the potential impact of the situation when making decisions.

| Type | Model | | | |
|------|-------|---|---|---|
|      | GPT-4 | LLaMA2-70B | GPT-3.5 | PaLM 2 |
| Standard | 31.2 | 27.2 | 34.7 | 14.8 |
| General | 36.6(+5.4) | 27.7(+0.5) | 36.1(+1.4) | 15.6(+0.8) |
| Specific | 46.5(+15.3) | 31.1(+3.9) | 44.1(+9.4) | 26.6(+11.8) |

Table 4: Performance of four models after incorporating general and specific prompts that remind the model of its existing knowledge. See detailed results for each type of representativeness heuristic in Table 12 and Table 13 of the Appendix. The relative performance of different prompt strategies with respect to standard prompting is indicated with script-size fonts.

For example, for Linda's problem, we assign a probability to each independent event to offer the model with more explicit statistical information. Nevertheless, the model has to infer the information based on the similarity manifested in the meaning of the statements in RHQs. The performance of the models on IQs is reported in Table 6 of the Appendix. Compared with SPQs' results (Table 5 of the Appendix), more errors were made for IQs. This further indicates that introducing scenarios interferes with the model's statistical decision-making process. This also explains why LLMs exhibit the representativeness heuristic more frequently when responding to RHQs.

## 6 Conclusion

We introduce a novel dataset (REHEAT) for the representativeness heuristic to assess whether models, like humans, make representativeness heuristic errors in decision-making—specifically, by overlooking known information and relying on similarity for judgment. Our research reveals that even the most advanced LLMs tend to repeat humans' heuristic mistakes when addressing issues related to the representativeness heuristic, highlighting the necessity for a deep understanding of these cognitive biases in data that then propagate into model decision-making processes. Furthermore, we explored how models perform differently when faced with various types of questions. For example, questions regarding the Conjunction Fallacy and Disjunction Fallacy present a significant challenge. This is due to the model's difficulty discerning the latent probabilistic relationships embedded within the text. We also found that hints designed to stimulate the model to recall its existing knowledge can, to some extent, enhance its performance. More specific and detailed prompts tend to lead the model to demonstrate better performance. However, although this method is effective, the model's potential is far from fully tapped, and there remains significant room for research exploration and improvement.

## Ethics Statement

The dataset involved in this work does not contain any sensitive information, such as personally identifiable information. It should be noted that although the output generated by the model could potentially contain harmful or biased information, based on our experimental observations, such situations have not occurred. We commit to continuously improving technology efficiency and ensuring our research promotes a fairer, safer, and more inclusive society.

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

# A Case Study

Here, we want to show some cases where the reasoning may be wrong, but the conclusion is surprisingly correct. These examples are very interesting and reflect some unique and exciting behavior patterns of the model when processing information.

## A.1 Counter-intuitive Question

Counter-intuitive question refers to a question that challenges common sense, expectations, or widely held beliefs. The following is an example of a counter-intuitive question; see Table 24:

*"In a city, most residents use bicycles to get around, while only a small percentage of people choose cars as their daily transportation. It is often assumed that car users are more concerned about the environment and healthy lifestyles, while bicycle users may be more concerned about comfort and convenience. If we learn that a particular resident is very concerned about the environment and health, is that person more likely to be an automobile user or a bicycle user? [1] automobile user, [2] bicycle user. "*

Generally speaking, in human perception, car users are more concerned about comfort and convenience, and bicycle users are more concerned about the environment and healthy lifestyles; the opposite statement is given in the question. This is a question of the Base Rate Fallacy type. Based on the description given in the question, `most residents use bicycles to get around, while only a small percentage of people choose cars as their daily transportation.` We know that most residents should use bicycles for daily commuting. The question asked `If we learn that a particular resident is very concerned about the environment and health, is that person more likely to be an automobile user or a bicycle user?` The answer based on base rate should be `[2] bicycle user,` and the answer based on similarity should be `[1] automobile user.`

Most of the models' answers are `[2] bicycle user,` which is correct. However, there are some interesting problems in reasoning. The model will ignore the counter-intuitive assumption given in the question and instead base it on the common-sense assumption that bicycle users are more concerned about the environment and healthy lifestyle (see the teal bold line in Table 24). Even under prompting like CoT, which can greatly improve reasoning accuracy, most models only mention the counter-intuitive assumption given in the question, and some do not mention it at all. They often consider this assumption incorrect and complete the reasoning based on common-sense assumptions. We hypothesize that this is because the training data is rich in examples and discourse that emphasizes cycling as an environmentally friendly and healthy mode of transportation. This common-sense assumption becomes the anchor for the model to answer questions, even when faced with a specific problem that sets conditions that contradict it.

This implies that, despite the significant improvements in LLMs' modeling and application capabilities, completely freeing them from the constraints of common-sense assumptions remains a challenge. Therefore, when designing and using these models, we need to consider how to balance the model's modeling of specific contexts and the application of general common sense.

## A.2 Sensitive Content Related Question (e.g. gender, race)

Sensitive content related questions, such as those about gender and race, when combined with the previously mentioned counter-intuitive questions, have led to some intriguing phenomena. Here, we will provide three examples (due to the length of the question, see Table 25, Table 26, and Table 27).

For example, in Table 25 and Table 26, gender-related content is mentioned, the `prevailing stereotype is that boys/girls are more likely to....` However, Table 25 does not contradict the common assumption, `boys are more likely to be involved in sports, while girls are likely to be more involved in the arts and theater,` whereas Table 26 is counter-intuitive assumption, `girls are more likely to be involved in sports,`

while boys are likely to be more involved in the arts and theater. Table 27, on the other hand, does not mention gender-related content but has counter-intuitive information. All three examples are Base Rate Fallacy type, so according to the base rate descriptions of the three questions, the answers to the three are, `[1] sports program`, `[2] arts and drama club`, and `[2] arts and drama club`. For Table 25 and Table 26, mentioning gender-related stereotypes makes models more vigilant, improving the accuracy of reasoning and conclusions. Especially for Table 26, unlike Table 27, the models do not default to common-sense assumptions when sensitive content is present.

We speculate that this phenomenon may originate from the model specifically programmed to recognize information related to sensitive content, thereby exhibiting exceptionally high sensitivity and vigilance when processing such information. This leads to increased correctness in reasoning and conclusions when the model deals with questions related to sensitive content. Since the model's attention to sensitive content can improve its reasoning and conclusion correctness when answering representative heuristics questions, this approach may be an effective way to enhance the model's accuracy in dealing with such questions.

## B  Limitation

This work is an initial attempt to introduce the representativeness heuristic into NLP and computational social science research, providing a new perspective for understanding the LLM behavior on the representativeness heuristic question. Nonetheless, the study has several significant limitations. Firstly, the exploration of representativeness heuristic is in its early stages, facing multiple challenges in practical applications. These include accurately quantifying and identifying representativeness heuristic behaviors and effectively utilizing these findings in different NLP tasks. Secondly, the datasets used in the existing research are relatively limited in size, and the human heuristic phenomenon encompasses not only the representativeness heuristic but also others, such as the anchoring heuristic and availability heuristic. We plan to add more questions designed around the representativeness heuristic in the future and gradually build datasets for other heuristics. Additionally, the evaluation of reasoning can be further refined, including understanding conceptual errors (such as misunderstanding Bayes' theorem), computational errors, and more. This paper evaluates the reasoning from a broad perspective. By adopting a more specific viewpoint, we might discover new insights into the behavior of LLMs. Another important limitation is that the current research primarily focuses on English datasets and models, limiting its global applicability and impact. The diversity of languages and cultures requires the verification and expansion of these findings across different linguistic and cultural backgrounds.

## C  Additional Tables

| Type | Model | | | |
|---|---|---|---|---|
| | GPT-3.5 | GPT-4 | LLaMA2-70B | PaLM 2 |
| Base Rate Fallacy | ✓ | ✓ | ✓ | ✓ |
| Conjunction Fallacy | ✓ | ✓ | ✓ | ✓ |
| Disjunction Fallacy | ✓ | ✓ | ✓ | ✓ |
| Insensitivity to Sample Size | ✓ | ✓ | ✓ | ✓ |
| Misconceptions of Chance | ✗ | ✓ | ✓ | ✓ |
| Regression Fallacy | ✓ | ✓ | ✓ | ✓ |

Table 5: Model performance on six Statistical Prototype Questions resulting from five iterations of self-consistency.

| Type | Model | | | |
|---|---|---|---|---|
| | GPT-3.5 | GPT-4 | LLaMA2-70B | PaLM 2 |
| Base Rate Fallacy | ✗ | ✓ | ✗ | ✓ |
| Conjunction Fallacy | ✗ | ✗ | ✗ | ✗ |
| Disjunction Fallacy | ✓ | ✓ | ✗ | ✗ |
| Insensitivity to Sample Size | ✓ | ✓ | ✓ | ✗ |
| Misconceptions of Chance | ✗ | ✓ | ✗ | ✓ |
| Regression Fallacy | ✓ | ✓ | ✓ | ✓ |

Table 6: Model performance on Intermediate Questions resulting from five iterations of self-consistency.

| Type | Strategy | | | |
|---|---|---|---|---|
| | Standard | Zero-shot CoT | Self-Consistency | One-Shot |
| Base Rate Fallacy | 31.9 | 38.3 (+6.4) | 36.2 (+4.3) | 26.1 (-5.8) |
| Conjunction Fallacy | 31.1 | 33.3 (+2.2) | 33.3 (+2.2) | 15.9 (-15.2) |
| Disjunction Fallacy | 22.9 | 18.8 (-4.1) | 22.9 (=) | 40.4 (+17.5) |
| Insensitivity to Sample Size | 45.1 | 25.5 (-19.6) | 43.1 (-2.0) | 44.0 (-1.1) |
| Misconceptions of Chance | 50.0 | 25.0 (-25.0) | 50.0 (=) | N/A (N/A) |
| Regression Fallacy | 71.4 | 85.7 (+14.3) | 71.4 (=) | N/A (N/A) |

Table 7: GPT-3.5 performance on REHEAT. The One-Shot approach was not evaluated on the 'Misconceptions of Chance' and 'Regression Fallacy' questions, due to their limited quantity. This limited quantity would also lead to significant fluctuations in accuracy changes for these question types. The relative performance of different prompt strategies with respect to standard prompting is indicated with script-size fonts.

| Type | Strategy | | | |
|---|---|---|---|---|
| | Standard | Zero-shot CoT | Self-Consistency | One-Shot |
| Base Rate Fallacy | 34.0 | 27.7 (-6.3) | 36.2 (+2.2) | 37.0 (+3.0) |
| Conjunction Fallacy | 20.0 | 26.7 (+6.7) | 24.4 (+4.4) | 27.3 (+7.3) |
| Disjunction Fallacy | 6.3 | 10.4 (+4.1) | 10.4 (+4.1) | 4.3 (-2.0) |
| Insensitivity to Sample Size | 54.9 | 64.7 (+9.8) | 58.8 (+3.9) | 74.0 (+19.1) |
| Misconceptions of Chance | 50.0 | 50.0 (=) | 50.0 (=) | N/A (N/A) |
| Regression Fallacy | 71.4 | 85.7 (+14.3) | 57.1 (-14.3) | N/A (N/A) |

Table 8: GPT-4 performance on REHEAT. The One-Shot approach was not evaluated on the 'Misconceptions of Chance' and 'Regression Fallacy' questions, due to their limited quantity. This limited quantity would also lead to significant fluctuations in accuracy changes for these question types. The relative performance of different prompt strategies with respect to standard prompting is indicated with script-size fonts.

| Type | Strategy | | | |
|---|---|---|---|---|
| | Standard | Zero-shot CoT | Self-Consistency | One-Shot |
| Base Rate Fallacy | 23.4 | 31.9 (+8.5) | 29.8 (+6.4) | 21.7 (-1.7) |
| Conjunction Fallacy | 20.0 | 31.1 (+11.1) | 24.4 (+4.4) | 27.3 (+7.3) |
| Disjunction Fallacy | 10.4 | 12.5 (+2.1) | 10.4 (=) | 63.8 (+53.4) |
| Insensitivity to Sample Size | 43.1 | 35.3 (-7.8) | 43.1 (=) | 52.0 (+8.9) |
| Misconceptions of Chance | 75.0 | 50.0 (-25.0) | 50.0 (-25.0) | N/A (N/A) |
| Regression Fallacy | 71.4 | 57.1 (-14.3) | 71.4 (=) | N/A (N/A) |

Table 9: LLaMA2-70B performance on REHEAT. The One-Shot approach was not evaluated on the 'Misconceptions of Chance' and 'Regression Fallacy' questions, due to their limited quantity. This limited quantity would also lead to significant fluctuations in accuracy changes for these question types. The relative performance of different prompt strategies with respect to standard prompting is indicated with script-size fonts.

| Type | Strategy | | | |
|---|---|---|---|---|
| | Standard | Zero-shot CoT | Self-Consistency | One-Shot |
| Base Rate Fallacy | 19.1 | 17.0 (-2.1) | 21.3 (+2.2) | 23.9 (+4.8) |
| Conjunction Fallacy | 2.2 | 2.2 (=) | 2.2 (=) | 7.3 (+5.1) |
| Disjunction Fallacy | 2.3 | 4.2 (+1.9) | 4.2 (+1.9) | 14.6 (+12.3) |
| Insensitivity to Sample Size | 22.0 | 19.6 (-2.4) | 39.2 (+17.2) | 30.0 (+8.0) |
| Misconceptions of Chance | 50.0 | 50.0 (=) | 75.0 (+25.0) | N/A (N/A) |
| Regression Fallacy | 71.4 | 28.6 (-42.8) | 85.7 (+14.3) | N/A (N/A) |

Table 10: PaLM 2 performance on REHEAT. The One-Shot approach was not evaluated on the 'Misconceptions of Chance' and 'Regression Fallacy' questions, due to their limited quantity. The limited quantity would also lead to significant fluctuations in accuracy changes for these question types. The relative performance of different prompt strategies with respect to standard prompting is indicated with script-size fonts.

| Type | GPT-4 | | | | LLaMA2-70B | | | |
|---|---|---|---|---|---|---|---|---|
| | TT | TF | FT | FF | TT | TF | FT | FF |
| Base Rate Fallacy | 25.6 | 14.9 | 2.1 | 57.4 | 19.1 | 4.3 | 12.8 | 63.8 |
| Conjunction Fallacy | 26.7 | 0.0 | 0.0 | 73.3 | 17.8 | 0.0 | 13.3 | 68.9 |
| Disjunction Fallacy | 8.3 | 0.0 | 2.1 | 89.6 | 10.4 | 0.0 | 2.1 | 87.5 |
| Insensitivity to Sample Size | 64.7 | 3.9 | 0.0 | 31.4 | 29.4 | 0.0 | 5.9 | 64.7 |
| Misconceptions of Chance | 50.0 | 0.0 | 0.0 | 50.0 | 50.0 | 0.0 | 0.0 | 50.0 |
| Regression Fallacy | 85.7 | 0.0 | 0.0 | 14.3 | 57.1 | 0.0 | 0.0 | 42.9 |

Table 11: The detailed combination of reasoning and outcome correctness for GPT-4 and LLaMA2-70B on the REHEAT via CoT prompts.

| Type | GPT-4 | | | LLaMA2-70B | | |
|------|----------|---------|----------|----------|---------|----------|
| | Standard | General | Specific | Standard | General | Specific |
| Base Rate Fallacy | 34.0 | 44.7(+10.7) | 53.2(+19.2) | 23.4 | 23.4(=) | 14.9(-8.5) |
| Conjunction Fallacy | 20.0 | 20.0(=) | 40.0(+20.0) | 20.0 | 24.4(+4.4) | 31.1(+11.1) |
| Disjunction Fallacy | 6.3 | 10.4(+4.1) | 14.6(+8.3) | 10.4 | 14.6(+4.2) | 14.6(=) |
| Insensitivity to Sample Size | 54.9 | 62.7(+7.8) | 70.6(+15.7) | 43.1 | 37.3(-5.8) | 51.0(+7.9) |
| Misconceptions of Chance | 50.0 | 50.0(=) | 50.0(=) | 75.0 | 50.0(-25.0) | 50.0(=) |
| Regression Fallacy | 71.4 | 71.4(=) | 85.7(+14.3) | 71.4 | 85.7(+14.3) | 100.0(+28.6) |

Table 12: Performance of GPT-4 and LLaMA2-70B after incorporating general and specific prompts that remind the model of its existing knowledge. The limited number of questions on 'Misconceptions of Chance' and the 'Regression Fallacy' could lead to significant fluctuations in accuracy changes for these question types. The relative performance of different prompt strategies with respect to standard prompting is indicated with script-size fonts.

| Type | GPT-3.5 | | | PaLM 2 | | |
|------|----------|---------|----------|----------|---------|----------|
| | Standard | General | Specific | Standard | General | Specific |
| Base Rate Fallacy | 31.9 | 34.0(+2.1) | 42.6(+10.7) | 19.1 | 17.0(-2.1) | 25.5(+6.4) |
| Conjunction Fallacy | 31.1 | 24.4(-6.7) | 35.6(+4.5) | 2.2 | 2.2(=) | 4.4(+2.2) |
| Disjunction Fallacy | 22.9 | 37.5(+14.6) | 37.5(+14.6) | 2.3 | 4.3(+2.0) | 6.7(+4.4) |
| Insensitivity to Sample Size | 45.1 | 37.3(-7.8) | 51.0(+5.9) | 22.0 | 22.0(=) | 52.9(+30.9) |
| Misconceptions of Chance | 50.0 | 50.0(=) | 50.0(=) | 50.0 | 75.0(+25.0) | 50.0(-25.0) |
| Regression Fallacy | 71.4 | 100.0(+28.6) | 100.0(+28.6) | 71.4 | 85.7(+14.3) | 100.0(+28.6) |

Table 13: Performance of GPT-3.5 and PaLM 2 after incorporating general and specific prompts that remind the model of its existing knowledge. The limited number of questions on 'Misconceptions of Chance' and the 'Regression Fallacy' could lead to significant fluctuations in accuracy changes for these question types. The relative performance of different prompt strategies with respect to standard prompting is indicated with script-size fonts.

| TYPE | PROMPT |
|------|--------|
| Multi-Choice | For multiple-choice questions, choose the one that you think is best, even if you think more than one option will work, choose the one you think is best. The answer should be accompanied by the number I gave for each option. For example, when the options are [1] choice1, [2] choice2, [3] choice3, if [1] choice1 is the correct answer, the answer format is [1] choice1. |
| Ranking | For ranking questions, provide a ranking, and make distinctions even if you think multiple options may have the same position in the ranking. When displaying the answer, you must provide the number assigned to each option in the original question. For example, when the options are [1] choice1, [2] choice2, [3] choice3. If you think [2] ranks before [1], and [1] ranks before [3], the answer format is [2] choice2 > [1] choice1 > [3] choice3. |

Table 14: System messages for different types of query questions.

| TYPE | QUESTIONS |
|---|---|
| Base Rate Fallacy | The probability of an event occurring in a general population is 1 in 10,000 ($P(A) = \frac{1}{10,000}$). A test designed to detect this event has: A true positive rate (sensitivity) of 99% ($P(B|A) = 0.99$), meaning it will correctly identify the event 99% of the time when it occurs. A true negative rate (specificity) of 99% ($P(\neg B|\neg A) = 0.99$), meaning it will correctly identify the non-occurrence of the event 99% of the time when the event does not occur. An individual is randomly selected, and the test indicates that the event is present (positive result). The question is to calculate $P(A|B)$, the probability that the event truly occurs given that the test indicates it does. Is $P(A|B)$: [1] Greater than 50%, or [2] Less than 50%? |
| Conjunction Fallacy | Given two independent events A and B, where the probability of A is 0.8 and the probability of B is 0.2, compare which is greater: the probability of event B or the probability of events A and B occurring together. |
| Disjunction Fallacy | Given two main categories A and B with probabilities P(A)=0.7 and P(B)=0.2, and two subsets a (of A) and b (of B), P(a) > P(B), rank the probabilities P(A), P(B), P(a), and P(b) based on the given and inferred hierarchical structure of probabilities. |
| Insensitivity to Sample Size | Given two populations, A and B, where A generates an average of 45 events per period and B generates an average of 15 events per period, and each event has a binary outcome with a base probability of 0.5, over a long-term observation (1 year), which population is expected to show greater variability in the proportion of periods during which the frequency of one specific outcome exceeds 60% of the events? Choose the most probable answer: [1] Population A (the larger population). [2] Population B (the smaller population). [3] The variability in both populations is about the same (that is, within 5% of each other). |
| Misconceptions of Chance | Consider a sequence of 10 independent events, each with two possible outcomes: A or B. The outcomes are equally likely, with the probability of each outcome being 0.5. Given that the first 5 events have all resulted in outcome A, what is the most likely final distribution of outcomes A and B after all ten events have occurred? Choose the most probable answer from the following: [1] 5 As and 5 Bs, [2] 10 As and 0 Bs, [3] The probability for both distributions is the same. |
| Regression Fallacy | Given a sequence of performance measurements with an average value of X, and observing that recent measurements significantly exceed X, followed by a sudden drop to a value of Y below X, which of the following is the most statistically probable explanation for the sudden drop, assuming no external factors are known to have influenced this change? [1] A specific external factor negatively impacted this particular measurement. [2] A random fluctuation within the expected variability of performance measurements. [3] A systematic change occurred but is not immediately apparent from the data provided. [4] The measurement at Y is an outlier caused by an unaccounted-for error in measurement. |

Table 15: Statistical Prototype Questions.

| TYPE | QUESTIONS |
|---|---|
| Base Rate Fallacy | A disease has an incidence rate of 1 in 10,000 in the population. A test for the disease is 99% accurate, meaning it has a 99% chance of correctly identifying the disease if it is present (true positive rate) and a 99% chance of correctly identifying the absence of the disease if it is not present (true negative rate). An individual tests positive for the disease. Question: What is the probability that the individual actually has the disease? Is it [1] greater than 50 percent or [2] less than 50 percent? |
| Conjunction Fallacy | Bill is 34 years old. He is intelligent, but unimaginative, compulsive, and generally lifeless. In school, he was strong in mathematics but weak in social studies and humanities. The probability of Bill being a physician is 0.3; the probability of Bill being an architect is 0.19; the probability of Bill playing poker as a hobby is 0.2; the probability of Bill playing jazz as a hobby is 0.4; the probability of Bill surfing as a hobby is 0.36; the probability of Bill is a reporter is 0.25; the probability of Bill is an accountant is 0.8; the probability of Bill climbs mountains for a hobby is 0.35. We have seven statements below: [1] Bill is a physician who plays poker as a hobby. [2] Bill is an architect. [3] Bill plays jazz as a hobby. [4] Bill surfs as a hobby. [5] Bill is a reporter. [6] Bill is an accountant who plays jazz as a hobby. [7] Bill climbs mountains for a hobby. Ranked the seven statements' probability from high to low. |
| Disjunction Fallacy | Danielle is sensitive and introspective. In high school, she wrote poetry secretly. Did her military service as a teacher. Though beautiful, she has little social life, since she prefers to spend her time reading quietly at home rather than partying. The probability of her studying Humanities is 0.7, and the probability of her studying Natural science is 0.2; Literature is the Humanities subset, and Physics is the Natural science subset. Rank the probability of the below options. [1] Literature, [2] Humanities, [3] Physics, [4] Natural science. |
| Insensitivity to Sample Size | Approximately 45 babies are born in the large hospital while 15 babies are born in the small hospital. Half (50%) of all babies born in general are boys. However, the percentage changes from 1 day to another. For a 1-year period, each hospital recorded the days on which >60% of the babies born were boys. The question posed is: Which hospital do you think the ratio varies more from day to day? Just choose the most probable answer below: [1] The larger hospital. [2] The smaller hospital. [3] About the same (that is, within 5% of each other). |
| Misconceptions of Chance | Two equally matched soccer teams are currently playing 10 games, and one team has won the first 5 games. For each game, each team's win rate is 50%. Which is the most likely final score of the game? Choose the most probable answer. [1] 5:5, [2] 10:0, [3] is the same probability. |
| Regression Fallacy | A basketball player has an average score of X points per game over a season. The player scores significantly above this average in the last several games, reaching new personal bests. However, in the next game, the player's score drops well below their season average to Y points. Without additional specific information about training, injuries, or team dynamics, which of the following is the most statistically probable explanation for the sudden drop in performance? [1] The player was poorly trained for this particular game. [2] The player sustained an injury that affected their performance. [3] There was discord within the team that impacted the player's performance. [4] No specific reason; it's a normal fluctuation in performance. |

Table 16: Intermediate Questions

| TYPE | HINTS |
|---|---|
| General | Please answer this question based on your statistical probability knowledge. |
| Base Rate Fallacy | Please answer this question based on your statistical probability knowledge, the conditional probability of an event needs to take into account the overall base rate (or prior probability) of that event. |
| Conjunction Fallacy | Please answer this question based on your statistical probability knowledge, the probability of two specific conditions occurring simultaneously is not higher than that of one specific condition. |
| Disjunction Fallacy | Please answer this question based on your statistical probability knowledge, the probability of a subset of events cannot be higher than the probability of the entire event. |
| Insensitivity to Sample Size | Please answer this question based on your statistical probability knowledge, changes in small samples tend to have a more pronounced effect on the overall statistics than changes in large samples. |
| Misconceptions of Chance | Please answer this question based on your statistical probability knowledge, the probabilities of each independent event are not related to each other. |
| Regression Fallacy | Please answer this question based on your statistical probability knowledge, a natural relationship between two variables does not imply causation. |

Table 17: Hints we use to remind LLMs to use their existing knowledge.

| FIELD | DESCRIPTION |
|---|---|
| Query | A doctor performs a test that is 99% accurate, and you test positive for the disease. However, the incidence of the disease is 1/10,000. How do you think you will get the disease? Choose between the following two probabilities: [1] greater than 50 percent, [2] less than 50 percent. |
| Type | Base Rate Fallacy |
| Ground Truth | [2] less than 50 percent |
| Model Answer | [2] less than 50 percent. |

Table 18: Example of Base Rate Fallacy in REHEAT answer by GPT-3.5, Standard prompt

| FIELD | DESCRIPTION |
|---|---|
| Query | Bill is 34 years old. He is intelligent, but unimaginative, compulsive, and generally lifeless. In school, he was strong in mathematics but weak in social studies and humanities. We have eight statements below: [1] Bill is a physician who plays poker as a hobby. [2] Bill is an architect. [3] Bill is an accountant. [4] Bill plays jazz as a hobby. [5] Bill surfs as a hobby. [6] Bill is a reporter. [7] Bill is an accountant who plays jazz as a hobby. [8] Bill climbs mountains for a hobby. Ranked the eight statements associated with each description by the degree to which Bill resembles the typical member of that class. |
| Type | Conjunction Fallacy |
| Ground Truth | [3] Bill is an accountant. > [4] Bill plays jazz as a hobby. > [7] Bill is an accountant who plays jazz as a hobby. |
| Model Reasoning | Sure, let's break down each statement and consider how well it matches with Bill's description.
1. Bill is a physician who plays poker as a hobby: Being a physician requires high intelligence, which matches Bill's description. However, it also often requires a level of creativity and adaptability, which might not suit Bill's unimaginative and compulsive nature. Poker requires a good understanding of probability and strategy, which could align with Bill's strength in mathematics.
2. Bill is an architect: This profession often requires high creativity and imagination, which doesn't align with Bill's description.
3. Bill is an accountant: This profession often requires a high level of detail orientation, compulsiveness, and a strong understanding of mathematics, all of which align well with Bill's description.
4. Bill plays jazz as a hobby: Jazz often requires high creativity and improvisation, which doesn't align with Bill's unimaginative nature.
5. Bill surfs as a hobby: Surfing is a very physical and adventurous hobby, which doesn't seem to align with Bill's description of being generally lifeless.
6. Bill is a reporter: This profession often requires high social skills and a strong understanding of humanities, which doesn't align with Bill's academic strengths.
7. Bill is an accountant who plays jazz as a hobby: As mentioned before, being an accountant aligns well with Bill's description, but playing jazz does not.
8. Bill climbs mountains for a hobby: Mountain climbing is a very physical and adventurous hobby, which doesn't seem to align with Bill's description of being generally lifeless.
Now, you can rank these statements based on how well they align with Bill's description. |
| Model Answer | [3] Bill is an accountant > [1] Bill is a physician who plays poker as a hobby > [7] Bill is an accountant who plays jazz as a hobby > [2] Bill is an architect > [6] Bill is a reporter > [4] Bill plays jazz as a hobby > [5] Bill surfs as a hobby > [8] Bill climbs mountains for a hobby. |

Table 19: Example of Conjunction Fallacy in REHEAT answer by GPT-4, CoT prompt

| FIELD | DESCRIPTION |
|---|---|
| Query | It is quite possible for something to be included in more than one option listed. For example, suppose the list is of foodstuffs, and one option is "frozen foods," while another is "desserts". In this case, do not interpret "frozen foods" as "frozen foods excluding deserts," nor interpret "desserts" as "desserts which are not frozen"—"ice cream" qualifies as both "frozen food" and "dessert." Danielle is sensitive and introspective. In high school, she wrote poetry secretly. Did her military service as a teacher. Though beautiful, she has little social life, since she prefers to spend her time reading quietly at home rather than partying. What is she most likely to study? Choose one answer that follows: [1] Literature, [2] Humanities, [3] Physics, [4] Natural science. |
| Type | Disjunction Fallacy |
| Ground Truth | [2] Humanities |
| Model Reasoning | Based on the information provided, Danielle is most likely to study [1] Literature or [2] Humanities, as she has a background in writing poetry and teaching, and prefers to spend her time reading quietly at home. It's less likely that she would study [3] Physics or [4] Natural science, as there is no indication of her having an interest or background in these fields. Therefore, the best answer is either [1] Literature or [2] Humanities. Based on the information provided, Danielle is most likely to study [1] Literature. Danielle's passion for reading and her background in writing poetry suggests that she has a strong interest in literature and the humanities. Additionally, her preference for spending time alone reading rather than socializing suggests that she enjoys immersing herself in written works. While it's possible that she may also be interested in other fields such as humanities or natural sciences, her interests and personality traits make literature the most likely choice for her to study. Therefore, the answer is [1] Literature. Based on the information provided, Danielle is most likely to study [1] Literature. Danielle's passion for reading and her interest in poetry suggests that she has a strong appreciation for literature and writing. Additionally, her introspective nature and preference for spending time alone reading rather than partying also indicate that she may be drawn to studying literature. The fact that she taught in military service also suggests that she might have an interest in humanities, but given her personal preferences and interests, literature seems like the most fitting choice. Therefore, option [1] Literature is the best answer. |
| Model Answer | [1] Literature |

Table 20: Example of Disjunction Fallacy in REHEAT answer by LLaMA2-70B, Self-Consistency prompt

| FIELD | DESCRIPTION |
|---|---|
| Query | Suppose there are two restaurants, one a large chain restaurant and the other a small family-run restaurant. Each restaurant records the number of days in a year that customers complain. The large chain restaurant serves an average of 1,000 customers per day, while the small family-run restaurant serves about 50 customers per day. Each restaurant is generally considered to have a complaint rate of about 20% The question is: which restaurant is more likely to have recorded customer complaints up to 40% over the course of the year? Just choose the most probable answer below: [1] Large chain restaurant. [2] Small family-run restaurant. [3] Both restaurants are roughly the same (i.e., the difference between the two is within 5%). |
| Type | Insensitivity to Sample Size |
| Ground Truth | [2] Small family-run restaurant. |
| Model Reasoning | The answer is: [1] Large chain restaurant. The large chain restaurant serves an average of 1,000 customers per day, while the small family-run restaurant serves about 50 customers per day. Each restaurant is generally considered to have a complaint rate of about 20%. This means that the large chain restaurant is likely to have 200 complaints per day, while the small family-run restaurant is likely to have 10 complaints per day. To calculate the probability of each restaurant having customer complaints up to 40% over the course of the year, we can use the following formula: Probability = (Number of complaints / Total number of customers) * 100% For the large chain restaurant, the probability of having customer complaints up to 40% is: Probability = (200 / 1000) * 100% = 20% For the small family-run restaurant, the probability of having customer complaints up to 40% is: Probability = (10 / 50) * 100% = 20% As you can see, the probability of each restaurant having customer complaints up to 40% is the same. However, the large chain restaurant is more likely to have a higher number of complaints overall, because it serves more customers. Therefore, the large chain restaurant is more likely to have recorded customer complaints of up to 40% over the course of the year. |
| Model Answer | [1] Large chain restaurant |

Table 21: Example of Insensitivity to Sample Size in REHEAT answer by PaLM 2, One-Shot prompt

| FIELD | DESCRIPTION |
|---|---|
| Query | Consider a regular six-sided die with four green faces and two red faces. The die will be rolled 20 times and the sequence of greens (G) and reds (R) will be recorded. You are asked to select one sequence, from a set of three. Please check the sequence of greens and reds on which you prefer to bet. Choose the most probable answer. [1] RGRRR, [2] GRGRRR, [3] GRRRRR, [4] Same probability. |
| Type | Misconceptions of Chance |
| Ground Truth | [1] RGRRR |
| Model Reasoning | Sure, let's break this down. First, let's understand the probability of rolling a green or a red. Since there are 4 green faces and 2 red faces, the probability of rolling a green (G) is $4/6 = 2/3$, and the probability of rolling a red (R) is $2/6 = 1/3$. Now, let's look at each sequence: 1. RGRRR: This sequence has 1 green and 4 reds. The probability of this sequence happening is $(2/3)^1 * (1/3)^4$. 2. GRGRRR: This sequence has 2 greens and 4 reds. The probability of this sequence happening is $(2/3)^2 * (1/3)^4$. 3. GRRRRR: This sequence has 1 green and 5 reds. The probability of this sequence happening is $(2/3)^1 * (1/3)^5$. We can see that the sequence with the most greens, which is more likely to be rolled, will be the most probable sequence. |
| Model Answer | [2] GRGRRR |

Table 22: Example of Misconceptions of Chance in REHEAT answer by GPT-3.5, Standard prompt

| FIELD | DESCRIPTION |
|---|---|
| Query | A basketball player scores higher than before in several games. In the next game, his score gets low. What might be the reason? Choose the reason from below: [1] poorly trained. [2] injury. [3] Discord within the team. [4] No specific reason, normal fluctuation. Choose the most possible answer. |
| Type | Regression Fallacy |
| Ground Truth | [4] No specific reason, normal fluctuation. |
| Model Reasoning | First, let's consider each option: 1. Poorly trained: This seems unlikely because the player has been scoring higher than before in several games. This indicates that the player is well-trained and capable of performing well. 2. Injury: This could be a possible reason. An injury could affect the player's performance. However, we don't have any information about the player being injured. 3. Discord within the team: This could also be a possible reason. Team dynamics can affect a player's performance. But again, we don't have any information about any discord within the team. 4. No specific reason, normal fluctuation: This is also a possible reason. It's normal for a player's performance to fluctuate from game to game. Now, let's rank these options based on the information we have. |
| Model Answer | [4] No specific reason, normal fluctuation. |

Table 23: Example of Regression Fallacy in REHEAT answer by GPT-4, CoT prompt

| FIELD | DESCRIPTION |
|---|---|
| Query | In a city, most residents use bicycles to get around, while only a small percentage of people choose cars as their daily transportation. **It is often assumed that car users are more concerned about the environment and healthy lifestyles, while bicycle users may be more concerned about comfort and convenience.** If we learn that a particular resident is very concerned about the environment and health, is that person more likely to be an automobile user or a bicycle user? [1] automobile user, [2] bicycle user. |
| Type | Base Rate Fallacy |
| Ground Truth | [2] bicycle user |
| Model Reasoning | To determine whether a person who is very concerned about the environment and health is more likely to be an automobile user or a bicycle user, we can consider the following factors: 
 n1. Environmental impact: **Bicycles are generally considered to have a lower environmental impact compared to cars, as they do not emit greenhouse gases or contribute to air pollution. This suggests that someone who is highly concerned about the environment may be more inclined to use a bicycle.** 
 n2. Health benefits: **Cycling is a form of exercise that promotes physical fitness and contributes to a healthy lifestyle. If someone prioritizes their health and well-being, they may be more likely to choose a bicycle as their mode of transportation.** 
 n3. City infrastructure: The availability of bike lanes, cycling infrastructure, and the overall bike-friendliness of the city can also influence the choice between a car and a bicycle. If the city has a well-developed cycling infrastructure, it may encourage more people to choose bicycles, regardless of their concerns about the environment and health. 
 **Considering these factors, it is reasonable to assume that a person who is very concerned about the environment and health is more likely to be a bicycle user rather than an automobile user.** However, it is important to note that individual preferences and circumstances can vary, and there may be exceptions to this general assumption. |
| Model Answer | [2] bicycle user |

Table 24: Case study example for A.1 in REHEAT, sentences highlighted in teal are the ones the case study focuses on.

| FIELD | DESCRIPTION |
|---|---|
| Query | In a diverse high school, most students participate in the school's sports program, and only a few participate in the school's arts and drama clubs. **The prevailing stereotype is that boys are more likely to be involved in sports, while girls are likely to be more involved in the arts and theater.** If we see a girl, is that student more likely to be a student in a sports program or a member of an arts and drama club? [1] sports program, [2] arts and drama club. |
| Type | Base Rate Fallacy |
| Ground Truth | [1] sports program |
| Model Reasoning | Sure, let's break this down:
1. The information given states that most students participate in the school's sports program. This means that the majority of the student body, regardless of gender, is involved in sports.
2. It is also mentioned that only a few participate in the school's arts and drama clubs. This indicates that the number of students involved in arts and drama is significantly less than those involved in sports.
3. The stereotype is that boys are more likely to be involved in sports, while girls are likely to be more involved in the arts and theater. **However, this is a stereotype and not a fact. It does not necessarily reflect the actual distribution of students in the school.**
4. Given these points, when we see a girl, we have two options to consider: [1] sports program, [2] arts and drama club.
Now, based on the information provided, you can make an informed decision. |
| Model Answer | [1] sports program. |

Table 25: Case study example for A.2 in REHEAT, sentences highlighted in teal are the ones the case study focuses on.

| FIELD | DESCRIPTION |
|---|---|
| Query | In a diverse high school, most students participate in the school's arts and drama clubs, and only a few participate in the school's sports program. **The prevailing stereotype is that girls are more likely to be involved in sports, while boys are likely to be more involved in the arts and theater.** If we see a girl, is that girl more likely to be a student in a sports program or a member of an arts and drama club? [1] sports program, [2] arts and drama club. |
| Type | Base Rate Fallacy |
| Ground Truth | [2] arts and drama club |
| Model Reasoning | Sure, let's break it down: 1. The information given states that most students in the school participate in the arts and drama clubs. This means that regardless of gender, the majority of students are involved in these clubs. 2. It's also mentioned that only a few students participate in the school's sports program. This indicates that the sports program has less participation overall. 3. The stereotype mentioned is that girls are more likely to be involved in sports, while boys are more likely to be involved in the arts and theater. **However, this is a stereotype and may not necessarily reflect the actual situation in the school.** 4. Given these points, when we see a girl in this school, we have to consider the overall participation rates in the sports program and the arts and drama clubs. Now, based on the information provided, you can make an informed decision. |
| Model Answer | [2] arts and drama club. |

Table 26: Case study example for A.2 in REHEAT, sentences highlighted in teal are the ones the case study focuses on.

| FIELD | DESCRIPTION |
|---|---|
| Query | In a diverse high school, most students participate in the school's arts and drama clubs, and only a few participate in the school's sports program. **The prevailing stereotype is that students who love reading are more likely to be involved in sports, while students who love sports are likely to be more involved in the arts and theater.** If we see a student who likes to read books, is that student more likely to be a student in a sports program or a member of an arts and drama club? [1] sports program, [2] arts and drama club. |
| Type | Base Rate Fallacy |
| Ground Truth | [2] arts and drama club |
| Model Reasoning | Sure, let's break it down:
1. The school is diverse and most students participate in arts and drama clubs, while only a few participate in the sports program.
2. The stereotype is that students who love reading are more likely to be involved in sports, and those who love sports are more likely to be involved in arts and drama.
3. We have a student who likes to read books.
Given these points, we might initially think that, according to the stereotype, a student who loves reading would be more likely to be involved in sports. **However, we know that students who love reading may fit more with arts and drama clubs.**
**So, even though the stereotype suggests that a book-loving student might be more likely to be involved in sports, the fact that the student is more likely to be involved in arts and drama clubs than in the sports program.**
Now, based on the information provided, you can make an informed decision. |
| Model Answer | [2] arts and drama club. |

Table 27: Case study example for A.2 in REHEAT, sentences highlighted in teal are the ones the case study focuses on.

