# OpenReview forum: "Will the Real Linda Please Stand up...to Large Language Models? Examining the Representativeness Heuristic in LLMs"
_colmweb.org/COLM/2024/Conference — COLM_

### Official Review · Reviewer_4fxR · 2024-04-25

**Rating:** 8
**Confidence:** 4
**Ethics Flag:** 1

**Summary:**

The authors create a new dataset, ReHeAT, that assesses susceptibility to the representativeness heuristic (RH), where humans judge scenarios similar to an exemplar to be more likely. The authors show that many popular LLMs exhibit RH biases, and that this bias can be improved by nudging the models towards using its in-weights knowledge. The paper is clear and polished, and the experiments are thoughtful and extensive.

**Questions To Authors:**

Comments:
- Change Subtitle 5.3: "Improving Performance by Hinting (that LLMs to ?) use Their Knowledge"
- Discussing your results in light of human base rates would be interesting. I see the note on page 8 that human base rates are around 10 to 50%. In a way, should we expect LLMs to exhibit this bias, in that they are trained to replicate the judgments of a population of humans?

**Reasons To Accept:**

- New benchmark for assessing language models for a known bias in humans.
- Concrete takeaways for the reader to fix this bias in their models.
- Thoughtful core premise supported by logical and interesting experiments.

**Reasons To Reject:**

I recommend acceptance.

---

> ### Author Rebuttal · Authors · 2024-05-30
>
> Dear reviewer 4fxrR,
>
> Thank you for your constructive comments and for recognizing the overall value of our work.
>
> - *Change Subtitle 5.3: “Improving Performance by Hinting (that LLMs to ?) use Their Knowledge”*
>
> **We have changed Subtitle 5.3 accordingly to “Improving Performance by Hinting that LLMs Use Their Knowledge.”**
>
> - *Discussing your results in light of human base rates would be interesting.*
>
> After receiving IRB approval from our institution, we have completed human evaluations on our proposed ReHeAT dataset. Through the Prolific platform, we recruited 153 participants to answer questions. Participants are asked to answer 3-4 types of representative heuristic questions. Human results are close to those of LLaMA2-70B, whereas the GPT series performs better than the human results. **We will incorporate these new results into our final version and provide more comprehensive outcomes.**
>
> | Type | Base Rate Fallacy | Conjunction Fallacy | Disjunction Fallacy | Insensitivity to Sample Size | Misconceptions of Chance | Regression Fallacy | Average |
> | --- | --- | --- | --- | --- | --- | --- | --- |
> | Human | 27.7 | 24.4 | 12.5 | 33.3 | 50.0 | 85.7 | 27.2 |
> | GPT-3.5 | 31.9 | 31.1 | 22.9 | 45.1 | 50.0 | 71.4 | 34.7 |
> | GPT-4 | 34.0 | 20.0 | 6.3 | 54.9 | 50.0 | 71.4 | 31.2 |
> | PaLM 2 | 19.1 | 2.2 | 2.3 | 22.0 | 50.0 | 71.4 | 14.8 |
> | LLaMA2-70B | 23.4 | 20.0 | 10.4 | 43.1 | 75.0 | 71.4 | 27.2 |
> - *I see the note on page 8 that human base rates are around 10 to 50%. In a way, should we expect LLMs to exhibit this bias, in that they are trained to replicate the judgments of a population of humans?*
>
> Thank you for asking this great question. **We do observe similar performance between humans and LMs. However, regarding whether we should expect LLMs to exhibit the representativeness heuristic, we think it depends on the LMs’ potential use.** When LLMs are used to aid in decision-making, we certainly do not want them to display such biases, as this cognitive trap can lead to incorrect judgments. From another viewpoint, this heuristic is a simplification strategy that helps people make quick decisions, which can be beneficial. If our goal is for LLMs to mimic human behavior closely, we should expect them to exhibit the representativeness heuristic. In reality, however, our aim extends beyond merely replicating human-like behavior; we also hope that LLMs will surpass human performance in areas prone to bias and other forms of error.

---

### Official Review · Reviewer_eSG3 · 2024-05-09

**Rating:** 8
**Confidence:** 3
**Ethics Flag:** 1

**Summary:**

This paper evaluates the impact of the representativeness heuristic on LLMs' reasoning. It proposes a dataset that covers various types of representativeness heuristic. Experiments show that LLMs make representativeness heuristic errors similar to humans and they behave differently when faced with different types of questions. In addition, providing hints in the prompts improves models' performance. The paper highlights the importance of considering the representativeness heuristic when using LMs for reasoning and decision-making.

**Questions To Authors:**

- The experiments on IQs and SPQs appear to only use one example per question type, but it remains unclear whether this performance would remain consistent with more examples. It would be better if the authors could include results using more examples.
- How are the ICL examples selected in section 5.1?
- Missing reference that compares human and LLMs reasoning, e.g. [1]

[1] Ishita Dasgupta, Andrew K. Lampinen, Stephanie C. Y. Chan, Hannah R. Sheahan, Antonia Creswell, Dharshan Kumaran, James L. McClelland, Felix Hill. Language models show human-like content effects on reasoning tasks.


## Comments after Author Discussion
Rating: 7 → 8

**Reasons To Accept:**

- Representativeness heuristic on LLMs is less explored in prior work, but understanding its impact on LLMs’ reasoning is important.
- This paper is well-written and easy to follow. The question types and experimental setup are clearly described. Examples of each type of questions are detailed in the appendix.
- The experiments are well-designed, especially the controlled ones on statistical SPQs and IQs. The qualitative analysis is comprehensive and interesting.

**Reasons To Reject:**

- This paper aims to investigate whether LLMs make representativeness heuristic mistakes similar to humans, but understanding the results is challenging without human performance for comparison. It would be helpful if the authors could conduct similar human studies or, at the very least, provide references from previous studies, given that some questions are directly drawn from prior research.
- The paper mentions “49 questions drawn directly from previous research and 153 new questions”, but the curation process for the new data is unclear. Also, how diverse are the new questions?

---

> ### Author Rebuttal · Authors · 2024-05-30
>
> Dear reviewer eSG3,
>
> We sincerely thank your insightful comments and valuable suggestions. Below, we address the important concerns you raised:
>
> - *It would be helpful if the authors could conduct similar human studies.*
>
> After receiving IRB approval from our institution, we have completed human evaluations on our proposed ReHeAT dataset. Through the Prolific platform, we recruited 153 participants to answer questions. Participants are asked to answer 3-4 types of representative heuristic questions. **We will incorporate these new results into our final version and provide more comprehensive outcomes.**
>
> | Type | Base Rate Fallacy | Conjunction Fallacy | Disjunction Fallacy | Insensitivity to Sample Size | Misconceptions of Chance | Regression Fallacy | Average |
> | --- | --- | --- | --- | --- | --- | --- | --- |
> | Human | 27.7 | 24.4 | 12.5 | 33.3 | 50.0 | 85.7 | 27.2 |
> | GPT-3.5 | 31.9 | 31.1 | 22.9 | 45.1 | 50.0 | 71.4 | 34.7 |
> | GPT-4 | 34.0 | 20.0 | 6.3 | 54.9 | 50.0 | 71.4 | 31.2 |
> | PaLM 2 | 19.1 | 2.2 | 2.3 | 22.0 | 50.0 | 71.4 | 14.8 |
> | LLaMA2-70B | 23.4 | 20.0 | 10.4 | 43.1 | 75.0 | 71.4 | 27.2 |
> - *The curation process for the new data is unclear. How diverse are the new questions?*
>
> The 153 new questions were organized to retain the essence of the representativeness heuristic being tested, at the same time changing the situations and contexts of the original items. Because each type of representativeness heuristic question shares the same statistical prototype, we aimed to display diversity in the contexts of the questions. There are approximately 15 to 18 different contexts for each type of question.
>
> - *It would be better if the authors could include results using more examples for SPQs and IQs.*
>
> Thank you for this advice. Each type of representativeness heuristic has the same statistical prototype. Therefore, only one SPQ is needed for each type, and we collected the majority of answers for each SPQ through self-consistency. Although IQs could indeed benefit from more examples for demonstration, considering that one of the purposes of IQs is to directly compare with SPQs, we adopted the same test format as the SPQs.
>
> - *How are the ICL examples selected in section 5.1?*
>
> **The examples in Section 5.1 for IC are randomly selected from diverse backgrounds.**
>
> - *Missing reference that compares human and LLMs reasoning, e.g. [1]*
>
> Thank you for recommending the missing references. **We will add this reference in our final version.**

---

### Official Review · Reviewer_f6wA · 2024-05-10

**Rating:** 8
**Confidence:** 4
**Ethics Flag:** 1

**Summary:**

This paper investigates an interesting topic, i.e. whether LLMs show typical representativeness heuristic as in human decision-making process.  The authors construct a test set covering six types of representativeness heuristics within the field of psychology, and evaluate multiple existing LLMs  including GPT-3.5, GPT-4, PaLM 2 and LLaMA 2.  Extensive experiments show that all LLMs demonstrated similar representativeness heuristic bias to humans, and advanced prompt techniques such as COT, self-consistency can only bring marginal improvements.

Overall , I think this is a good paper.  Studying and understanding what kind of shortcoming LLMs have inherited from texts produced by human is important to build intelligent systems that are smarter than human, and I believe this paper takes an important step towards this goal.  The topic also perfectly fits the theme of COLM.

The paper also shines in its strong experiments, discussing multiple prompting techniques to improve models' performance on the proposed REHEAT dataset.

**Questions To Authors:**

- Do you think more sophistic prompting strategies such as multi-agent debating can bring larger improvements?
- It would be better to provide experiments of testing models performance across different scales.
- Is there any results on average human accuracy on the proposed testset?

**Reasons To Accept:**

- important research topic
- thorough experimental results
- a newly constructed datasets

**Reasons To Reject:**

- no obvious shortcomings

---

> ### Author Rebuttal · Authors · 2024-05-30
>
> Dear reviewer f6wA,
>
> Thank you for your positive and constructive feedback. Below, we respond to your questions.
>
> - *Do you think more sophistic prompting strategies such as multi-agent debating can bring larger improvements?*
>
> **We believe more sophisticated prompting strategies could bring improvements, similar to how humans, when faced with heuristic questions, think deeply, and invest more mental resources to avoid heuristic biases.** We will explore more complex prompting strategies in our future work.
>
> - *It would be better to provide experiments of testing models performance across different scales.*
>
> Thank you for your advice. **We provide additional experiments on LLaMA2-13B to compare with LLaMA2-70B.**
> The performance of LLaMA2-13B is slightly lower than that of LLaMA2-70B. We can infer that model size impacts performance on representativeness heuristic questions, with larger models generally enhancing reasoning capabilities to a certain extent.
>
> | Model | Standard | Zero-shot CoT | Self-Consistency | One-Shot ICL |
> | --- | --- | --- | --- | --- |
> | LLaMA2-13B | 24.3 | 28.2 (+3.9) | 23.8 (-0.5) | 41.1 (+16.8) |
> | LLaMA2-70B | 27.2 | 29.2 (+2.0)  | 29.2 (+2.0) | 41.7 (+14.5) |
> - *Is there any results on average human accuracy on the proposed testset?*
>
> After receiving IRB approval from our institution, we have completed human evaluations on our proposed ReHeAT dataset. Through the Prolific platform, we recruited 153 participants to answer questions. Participants are asked to answer 3-4 types of representative heuristic questions. Human results are close to those of LLaMA2-70B, whereas the GPT series performs better than the human results. **We will incorporate these new results into our final version and provide a more comprehensive analysis.**
>
> | Type | Base Rate Fallacy | Conjunction Fallacy | Disjunction Fallacy | Insensitivity to Sample Size | Misconceptions of Chance | Regression Fallacy | Average |
> | --- | --- | --- | --- | --- | --- | --- | --- |
> | Human | 27.7 | 24.4 | 12.5 | 33.3 | 50.0 | 85.7 | 27.2 |
> | GPT-3.5 | 31.9 | 31.1 | 22.9 | 45.1 | 50.0 | 71.4 | 34.7 |
> | GPT-4 | 34.0 | 20.0 | 6.3 | 54.9 | 50.0 | 71.4 | 31.2 |
> | PaLM 2 | 19.1 | 2.2 | 2.3 | 22.0 | 50.0 | 71.4 | 14.8 |
> | LLaMA2-70B | 23.4 | 20.0 | 10.4 | 43.1 | 75.0 | 71.4 | 27.2 |
>
> *Note.* Here we compare Human results with the LMs’ Standard results of all other models.

---

### Official Review · Reviewer_3FvA · 2024-05-11

**Rating:** 8
**Confidence:** 3
**Ethics Flag:** 1

**Summary:**

This paper presents an evaluation suite testing language models for the "representativeness heuristic" cognitive bias. It reviews six types of this heuristic, with illustrations, detailed descriptions, and examples, and reviews the results of testing LMs for these biases. The paper includes results for:
* Four LLMs,
* four prompting approaches,
* varying numbers of few-shot examples,
* correctness of the models' reasoning,
* two methods of 'hinting' the models, and
* two easier sets of related questions ("SPQs" and "IQs") testing necessary knowledge for solving the problems.

These experiments show that LMs exhibit qualitatively similar behavior to humans on such questions.

**Questions To Authors:**

* More of a comment: the base rate fallacy picture doesn't seem to be a great example of the fallacy. Would be nice if you could illustrate it there too. Unless I'm missing something?
* The caption on the Disjunction Fallacy image is ungrammatical.

**Reasons To Accept:**

This is an interesting and well-executed study of LM reasoning. The paper is super well written and clear, focusing clearly on a question and testing it thoroughly. Lots of extra details are given in the appendix. This is probably a good evaluation to add to the long running list of interesting LM evaluations to run on frontier models. Lots of different settings are tried and tested and results are reported super clearly and legibly. I appreciate the text coloring and sizing in the tables. Overall just a nice paper.

**Reasons To Reject:**

Definitely not worth rejecting over, but I think what would really improve this paper a lot is spending some time discussing construct validity and external validity. Why are we testing models for these kinds of biases? What do we conclude about them when they get these results?

With humans, results on these tests may be taken to be indicative of broader reasoning patterns outside of the testing setting. But we already know that can be pretty problematic — psychology tests are run on a select group of people in a very artificial environment, subjects generally try to perform the way they think they are expected to perform, and the results may not be representative of how people act in the real world (See [this paper](https://www2.psych.ubc.ca/~schaller/528Readings/Rozin2001.pdf) for a nice discussion of this in the context of social psych). So there are important caveats to these kinds of results, but we take them seriously because of what we understand about the mental processes behind human decision making.

We do not have a comparable understanding of LM decision making, into which we do not have introspection, and which is much less consistent and systematic than human decision making. So what does it mean that LMs respond like humans do? Perhaps it just means they were trained on human cognitive bias tests, and any other place where they are presented with the opportunity to do such reasoning, they will behave very differently! Or perhaps these biases will show up in quite a different way when the LM is presented with a more complex or compound problem to solve, and its implicit reasoning won't follow the same patterns. It's really hard to say how these kinds of tests generalize to explain human behavior more broadly, and the case is much, much harder for LMs.

Not to say it's not worth doing and trying, but I think this is an important scientific consideration and a serious limitation of this kind of study, and I think the paper would be much better if it addressed it carefully.

---

> ### Author Rebuttal · Authors · 2024-05-30
>
> Dear reviewer 3FvA,
>
> Thank you for your encouraging, thorough, and positive feedback. Below, we address your comments.
>
> - *What would really improve this paper a lot is spending some time discussing construct validity and external validity. Why are we testing models for these kinds of biases? What do we conclude about them when they get these results?*
>
> **We agree that further discussing construct validity and external validity will improve this paper, and we will add further discussion to the final version.**
> To address construct validity, psychologists develop measures and apply psychometrics to measure individual differences in personal characteristics (e.g., traits, behaviors) as well as individual differences in intra-individual change (e.g., states, behavioral patterns). In a similar spirit of construct validity, here seek to measure the consistency between models, so that any empirical convergence provides indirect evidence of construct validity. Additionally, in psychology, common methods for improving external validity include improving the diversity of samples and improving the psychological fidelity of the data, so that both are more likely to generalize across situations. In this vein, when we explicitly attempt to provide diverse contexts for each type of representativeness heuristic and more realistic descriptions. **We test the model's performance on these diverse and high-fidelity questions in our dataset to achieve external validity.**
>
> - *The base rate fallacy picture doesn’t seem to be a great example of the fallacy. Would be nice if you could illustrate it there too. Unless I’m missing something?*
>
> Thank you for the advice regarding the base rate fallacy figure. We agree that although the current image reflects the Bayesian theorem of the base rate fallacy, it does not demonstrate its counterintuitive aspect. **We will change the figure in the following way:** (a) the blue dots within the large circle, but outside the small circle, will change to red, and (b) the red dots within the large circle, but outside the small circle, will change to blue. Revising in this manner shows how a high proportion of those who are sick have symptoms, but that does not mean that a high proportion of those with symptoms are sick.
>
> - *The caption on the Disjunction Fallacy image is ungrammatical.*
>
> **We will change it to** “A or B. A. People may think if A, then not B. However, if A is a subset of B, this conclusion would be incorrect.”

---

### Decision · Program_Chairs · 2024-07-10

**Decision:**

Accept

**Comment:**

This is a solid paper that brings a set of well-documented cognitive/social psychology phenomena related to a theoretically important idea in those fields, the representativeness heuristic, into the picture of studying and evaluating LLMs. There's great concurrence among reviewer evaluations and I agree that the paper is well executed and a valuable contribution. I recommend acceptance.

[At least one review was discounted during the decision process due to quality]